# Examining the psychosocial impacts of the COVID-19 pandemic: an international cross-sectional study protocol

Sandila Tanveer [1], Philip J Schluter [2,3] Richard J Porter,[1] Joseph Boden,[1] Ben Beaglehole,[1] Ruqayya Sulaiman-Hill [1], Shaystah Dean,[4] Romana Bell,[5] Wafaa N Al-Hussainni,[6] Maliheh Arshi,[7] Amer Siddiq Amer Nordin,[8] Mehmet Dinç,[9] Mussarat Jabeen Khan,[10] Mohammad Sabzi Khoshnami,[7] Muthana A Majid Al-Masoodi [11], Amir Moghanibashi-Mansourieh,[7] Sara Noruzi,[12] Anggi Rahajeng,[13] Shaista Shaikh,[14] Nisa Tanveer,[15] Feyza Topçu [9], Saadet Yapan,[9] Irfan Yunianto,[16] Lori A Zoellner,[17] Caroline Bell[1]

For numbered affiliations see end of article.

**Correspondence to**
Dr Caroline Bell;
caroline.bell@otago.ac.nz

## ABSTRACT

**Introduction** The COVID-19 pandemic exposed people to significant and prolonged stress. The psychosocial impacts of the pandemic have been well recognised and reported in high-income countries (HICs) but it is important to understand the unique challenges posed by COVID-19 in low- and middle-income countries (LMICs) where limited international comparisons have been undertaken. This protocol was therefore devised to study the psychosocial impacts of the COVID-19 pandemic in seven LMICs using scales that had been designed for or translated for this purpose.

**Methods and analysis** This cross-sectional study uses an online survey to administer a novel COVID Psychosocial Impacts Scale (CPIS) alongside established measures of psychological distress, post-traumatic stress, well-being and post-traumatic growth in the appropriate language. Participants will include adults aged 18 years and above, recruited from Indonesia, Iraq, Iran, Malaysia, Pakistan, Somalia and Turkey, with a pragmatic target sample size of 500 in each country.

Data will be analysed descriptively on sociodemographic and study variables. In addition, CPIS will be analysed psychometrically (for reliability and validity) to assess the suitability of use in a given context. Finally, within-subjects and between-subjects analyses will be carried out using multi-level mixed-effect models to examine associations between key sociodemographic and study variables.

**Ethics and dissemination** Ethical approval was granted by the Human Ethics Committee, University of Otago, New Zealand (Ref. No. 21/102). In addition, international collaborators obtained local authorisation or ethical approval in their respective host universities before data collection commenced.

Participants will give informed consent before taking part. Data will be collected and stored securely on the University of Otago, New Zealand Qualtrics platform using an auto-generated non-identifiable letter-number string. Data will be available on reasonable request. Findings will be disseminated by publications in scientific journals and/or conference presentations.

**Trial registration number** NCT05052333.

## STRENGTHS AND LIMITATIONS OF THIS STUDY

⇒ This cross-sectional observational study will make a significant contribution to understanding the psychosocial impacts of the COVID-19 pandemic in seven low- and middle-income countries with different pandemic impacts.

⇒ This study will quantify psychometric measures in countries where there is currently limited access to newly developed scales relating to COVID-19 and associated measures of psychological distress, well-being and post-traumatic growth in appropriate languages.

⇒ The findings will provide psychometric information on the novel COVID Psychosocial Impacts Scale (CPIS) in differing pandemic contexts.

⇒ There are limitations associated with selective recruitment and response rate in this study; which may limit the generalisability of the findings and comparability between groups.

## INTRODUCTION

COVID-19 was identified as an infectious disease by the World Health Organization (WHO) on 11 February 2020.[1] This new virus rapidly spread and was declared an international public health emergency on 30 January 2020 and a pandemic on 11 March 2020.[1] As of 31 January 2023, there have been 753 497 572 confirmed cases of COVID-19 including 6 813 301 deaths reported by WHO[2]; and likely many more as a result of under-reporting.[3] The COVID-19 pandemic has been complex in its origin, spread, impacts and consequences. Specifically, COVID-19 exposed people to significant and prolonged stress from often overlapping concerns like health consequences of COVID-19, impacts of public health restrictions (e.g., lockdowns,

social distancing, travel bans), financial losses, vaccination mandates, and associated societal divisions.[4]

When examining the psychosocial impacts in a pandemic context, researchers experience key challenges not only with the choice of study design and measures but also in providing a multicultural perspective.[5] In this respect, researchers have obtained data using convenience[6 7] or representative sampling[8] by employing either cross-sectional[8] or longitudinal[7 9] design. Those studies have been carried out either on the general population[6–8] or on potentially vulnerable populations (such as the health workforce[10–12] or those with pre-existing risk factors such as having a previous history of mental disorders[9 13 14]). Most studies include self-report measures of well-being, psychological distress, depression, anxiety and post-traumatic stress disorder (PTSD).[15] However, these standardised measures do not examine the particular impacts of COVID-19-related stressors and there has been a call for novel COVID-19-specific measures to be used alongside established psychometric scales.[16] In response, several COVID-19-specific scales have been developed which vary in focus and length. For example, one scale assesses fear associated with COVID-19 infection[17] (see also Akan[6]), while another one examined a wide range of impacts of the pandemic and public health restrictions on physical health, employment and finances, and family and social disruption.[18] Notably, the latter scale, in attempting to capture the wider psychosocial impacts, frequently reflects the context and country in which they were developed, limiting their more general applicability in other contexts. Researchers have also attempted to develop and validate brief screening tools for clinical research and practice; e.g., the 5-item Coronavirus Anxiety Scale[19] and the 7-item COVID-19 Anxiety Scale.[20] In addition, while the positive outcomes of the pandemic have been recognised,[21] few studies formally assessed the positive impacts,[22 23] which may limit a comprehensive understanding of the psychosocial impacts of COVID-19.

To date, the majority of the literature examining the psychosocial impacts of COVID-19 is from studies conducted in China, Europe and the USA and a few multinational studies have been carried out among the general population. A multinational study that examined the psychosocial impacts of COVID-19 across several high-income countries (HICs) found generalised anxiety among 21.0% and major depression among 25.5% of their respondents.[24] Another study that examined the psychological impacts of COVID-19 across several HICs and low- and middle-income countries (LMICs), reported high rates of anxiety (28.2%), stress (18.3%) and depression (26.6%) in their community-based samples[25] (see also[26] for depression prevalence in LMICs). Some multinational studies have reported significant differences in these psychological impacts across countries.[26 27] In comparison, studies investigating the psychosocial impacts in other regions, especially in LMICs are scarce, despite many of these areas being severely affected by COVID-19. The difficulties in evaluating the pandemic-related

psychological impacts in these regions are compounded by the lack of standardised measures in the local languages of these countries, the absence of which limits the comparison of results between countries.

To address the issues identified above, we developed the 50-item COVID Psychosocial Impacts Scale (CPIS) to comprehensively assess the impacts of the pandemic including the adverse personal, social and economic consequences that followed. The construction of the scale was informed by our team's experience assessing the psychosocial impacts of the Canterbury Earthquakes[28 29] and the Mosque Attacks in 2019.[30] Notably, the existing pandemic-related scales were also consulted in scale development.[6 16–20] The items and subscales of CPIS were modelled on the Social Readjustment Scale[31] assessing exposure to pandemic-related life events (disruptions to personal, family, employment, faith, education and daily routines) and the resultant level of stress. It also included questions examining the potential positive consequences of the COVID-19 pandemic. The CPIS was refined using the 'group mind' process,[32] with colleagues from a range of countries and cultures (Bangladesh, Ghana, Pakistan and Turkey) reviewing a draft of the questionnaire, with iterative improvements based on their comments. The applicability of the CPIS for use in a range of countries and cultures was duly considered through collaboration with international colleagues.

The 50-item CPIS has been validated in a non-representative New Zealand sample at two-time points (2020, 2022) when the population exposure to COVID-19 community infections varied markedly.[33] In that study, the CPIS was administered with a measure of psychological distress (Kessler Psychological Distress Scale, K10)[34] and well-being (WHO Well-Being Index, WHO-5).[35] Findings indicate a unidimensional structure within CPIS subscales while it correlates with psychological distress and general well-being is not only a proxy for these constructs.[33] Based on findings from the validation study, we found that some items were less important in future iterations of the CPIS. Therefore, we have refined CPIS from 50 to 32 items.

In addition to developing the CPIS, the research team took the opportunity to use resources that had already been deployed for the translation of the research instruments for the March 15th Project[30] to develop the CPIS into several different language versions (Arabic, Indonesian, Malay, Persian, Somali, Turkish and Urdu). To ensure that translations captured psychometric meanings in a culturally appropriate way, translations were carried out by proficient bilingual research assistants and interpreters who were familiar with the content and context, using parallel and back-to-back translations in an iterative process. These translations were then compared for internal consistency and examined for face validity.

Our planned programme of work is to examine the 32-item CPIS and translated psychometric measures in the countries of interest. This protocol describes a cross-sectional observational study to assess the psychosocial impacts of the COVID-19 pandemic in seven LMICs

(Indonesia, Iran, Iraq, Malaysia, Pakistan, Somalia and Turkey). It uses an online survey that includes the 32-item CPIS and standardised measures of psychological distress, post-traumatic stress, well-being and post-traumatic growth. Our findings will increase our understanding of the psychosocial impacts of the COVID-19 pandemic across several LMICs with differing rates of infection, morbidity and mortality. Findings will also provide psychometric evidence on the novel CPIS. As a result, our research will contribute to a growing body of mental health measures that can be used for cross-cultural comparison.

## METHODS AND ANALYSIS
### Study design
This is a cross-sectional observational study examining the psychosocial impacts of the COVID-19 pandemic being conducted in seven LMICs.

### Measures
The measures have all been translated into the appropriate language for the country in which they will be administered (see online supplemental file). They include the following.

#### Sociodemographic measures
Sociodemographic measures include age, gender, marital status, religion, ethnicity, self-reported English proficiency, highest level of education, occupation, study or work status and monthly income (in local currency). On the recommendation of the international collaborators, some of the demographic questions may vary to ensure contextual appropriateness. For example, in Turkey, students do not study part-time so that option was removed from the item.

#### Prior trauma exposure
The exposure to previous trauma will also be sought to examine the effect of previous trauma on the psychosocial impacts of COVID-19. Participants will be asked to select as many as appropriate from a list of previous exposure to natural disasters, war or military conflict, childhood adversity, physical or sexual assault, and serious physical accident.

#### COVID Psychosocial Impacts Scale
This is a novel 32-item measure used to examine the psychosocial impacts of the COVID-19 pandemic. The scale asks participants if they have experienced stressors in response to COVID-19 and then to indicate the magnitude of stress on a 6-point scale scored from 0='no', 1='yes, no stress at all' to 5='yes, a lot of stress'. An overall pandemic-related stress score can be calculated ranging from 0 to 160.

#### Kessler-10
This is a 10-item measure assessing symptoms of psychological distress in the previous 30 days.[34] Each item is scored from 1='none of the time' to 5='all of the time'.

The raw score can range from 10 to 50; with scores 30 and above indicating severe mental distress.[36]

#### WHO Well-Being Index
This is a short 5-item measure assessing subjective psychological well-being in the previous 2 weeks.[35] Each item is scored from 0='at no time' to 5='all of the time'. The raw score range is 0–25; this can be multiplied by 4 to give a final score, with 100 representing the highest subjective well-being rating.

#### PTSD Checklist for DSM-5
This is a 20-item measure assessing symptoms of PTSD in the past month anchored in response to COVID-19 as a potentially traumatic event. Each item is scored from 0='not at all' to 4='extremely'; with a severity score range of 0–80.[37]

#### Post-traumatic Growth Inventory
This is a 21-item measure assessing change following exposure to trauma,[38] adapted to reflect how change occurred in your life as a result of COVID-19. There are five subscales (relating to others, new possibilities, personal strength, spiritual change and appreciation of life). Each item is scored from 0='not at all' to 5='a lot' with a score range of 0–105, with higher scores reflecting positive transformation after the traumatic event.

### Participants
Adults (aged 18 years and over) who reside in Indonesia, Iran, Iraq, Malaysia, Pakistan, Somalia or Turkey will be recruited as participants. Anyone aged below 18 years and those not currently residing in the country of interest will be excluded from the study.

### Study status
Different sites are at different stages of project implementation including ethical approval pending (Iran); ethical approval obtained and recruiting participants (Somalia) and participant recruitment completed (Indonesia, Iraq, Malaysia, Pakistan and Turkey).

### Recruitment
The mode of recruitment will vary between countries, but will generally involve the distribution of the survey link using a participant mailing list or social media list held by each international collaborator at the host site. The local researchers will be using sampling techniques that are appropriate within their cultures and contexts, which for the most part resemble snowball sampling techniques.

### Procedure
Participants receive a survey link via the recruitment method above. The online survey comprises a participant information sheet, consent form and questionnaires. If a subject consents, they are asked to complete the online survey (which includes the self-report measures described above) using the Qualtrics XM online survey platform. The survey takes approximately 15–20 min to complete

and is provided in the appropriate language for the site. Information is provided about local supports that can be accessed if the self-report questions were to cause distress.

## Patient and public involvement

The project has been designed with community and general public involvement. Feedback has also been taken on project design from academics and the general public in the countries of interest.

## Data management

To ensure the anonymity of the obtained dataset the following standard operating procedures are in place. Where applicable, only the international collaborators hold contact information for their participant pool. Each site has specific guidelines to preserve anonymity specified in their local ethics application. Following collection, data will be saved on the University of Otago, New Zealand Qualtrics platform using an auto-generated non-identifiable letter-number string. Importantly, the obtained data set will be completely anonymous as no identifiable information is obtained in the survey.

## Data analysis

Incomplete survey responses will be removed from the analyses. Statistical analyses will be carried out to examine the psychosocial impacts of the COVID-19 pandemic in different countries and provide psychometric evidence using SPSS V.28.0[39] and Stata Release V.17.[40] Reporting will follow 'Strengthening the Reporting of Observational Studies in Epidemiology' guidelines.[41]

### Sample size estimation

As no prevalence of the psychosocial impacts of the COVID-19 pandemic in the general population of the countries of interest exists, no formal sample size calculations were conducted. The final target sample size of n=500 per country was pragmatically selected balancing the competing demands of maximising statistical power, expedited cost-effective data collection processes and international reach while simultaneously minimising institutional burden.[42]

### Psychometric analyses

Reliability analyses will be carried out on each measure to determine the suitability of its use, employing item-total correlations and Cronbach's α as a measure of internal homogeneity and consistency. The construct validity of CPIS with WHO-5 and K10 will be examined using a correlation matrix.

### Within-subjects analyses

Descriptive analysis will be carried out on sociodemographics (e.g., age, gender, ethnicity, religion, marital status, the highest level of education, work status, monthly household income, exposure to previous trauma) and study variables (well-being, psychological distress, post-traumatic stress, post-traumatic growth, pandemic-related stress from the CPIS) to inform site-specific sociodemographic profile and psychosocial outcomes. To explore the relationship between sociodemographic and study variables, follow-up analyses will be carried out using a linear mixed model (also known as multi-level modelling) to pool predictor variables. This will be carried out for each of the seven sites individually.

### Between-subjects analyses

To look at empirical distributions across countries, graphing superimposed Cumulative Distribution Function (cdf) will be used. In addition, to understand the role of demographic and study variables, multi-level modelling will be carried out in which participants will be nested within their countries—which will be treated as random effects to investigate the association between probable distress and well-being indicators and sociodemographic and pandemic-related variables. This approach allows individual and country-level factors to be included. Residual diagnostics and model assumptions will also be checked.

## DISCUSSION

This is a cross-sectional observational research that examines the psychosocial impacts of the COVID-19 pandemic across seven LMICs using a novel CPIS and measures of well-being, psychological distress, post-traumatic stress and post-traumatic growth. Despite LMICs being significantly impacted by COVID-19, few studies have been published from these regions. This study will therefore provide findings to address this important gap in the literature. The multi-centre design of the study will allow comparisons between the empirical distributions across countries. Furthermore, by including individual and country-level factors, multi-level modelling will enable us to understand the role of demographic and study variables. The findings have the potential to contribute to the understanding of the psychosocial impacts of the COVID-19 pandemic in countries that have not been well studied and to allow cross-cultural, multi-site comparisons.

Several multinational studies have been carried out examining the psychosocial impacts of COVID-19 among the general population using either standardised measures alone[11 24 25 27 43] or in combination with novel instruments.[26 44] These studies mostly obtained data from non-representative samples (except one[24]) from HICs,[24 45 46] LMICs[26] or across HICs and LMICs.[25 27 43 44] These studies have reported different rates in the prevalence of depression,[11 24–27 43] anxiety,[24 25 43] stress[11 25 27] and PTSD.[11] A few studies have examined the sociodemographic variables in relation to COVID-19, and have found an association of depression with younger age, gender (female), high levels of exposure and stigmatisation related to COVID-19.[26 45] Others have reported the association of stress with gender (female), marital status (single), no formal education level, religion, being exposed to a confirmed or suspected COVID-19 patient, and being forced to be quarantined.[44 46]

Against this background, our study will comprehensively examine the psychosocial impacts of COVID-19 using standardised measures in combination with novel CPIS across seven LMICs examining both sociodemographics and study variables. Our study has the potential to make several contributions to the existing literature. The 32-item CPIS is a novel scale that comprehensively assesses the psychosocial impacts of the pandemic, with the scale design allowing assessment of both the number of exposures to specific pandemic-related life events and the associated level of stress related to each of these events. The psychosocial impacts of the pandemic-related life events will be examined cross-culturally in this study, which distinguishes it from the existing multinational studies that mainly focus on the prevalence of mental health disorders and do not examine the adverse personal, social and economic consequences that followed COVID-19 pandemic. Importantly, the multilingual nature of the study is an important contribution. An underlying factor to the lack of studies in LMIC has been the lack of availability of measures translated into appropriate languages, which this study addresses. Hence, our findings will contribute to a growing body of mental health measures that can be used for cross-cultural comparison.

In addition, the study will also correlate outcomes on the CPIS with other validated measures of well-being and psychological distress and provide essential psychometric information on scale reliability and validity. Although, using pandemic-related scales in combination with standardised measures has been adopted by a few studies,[26 44] no formal psychometric analyses of the measures' suitability in varying contexts were undertaken. Additionally, to date, only a few nationwide studies have formally assessed potential positive outcomes associated with the pandemic.[21–23] The findings from this study will therefore contribute for the first time to this developing literature of examining the positive impacts of the COVID-19 pandemic in a cross-cultural context.

There are, however, limitations associated with the proposed study design. This study makes use of a novel CPIS developed during the COVID-19 pandemic and a cross-sectional design. As a consequence, there is no prepandemic comparison data. Repeated use of the CPIS over time and during different phases of the pandemic is needed to inform whether the CPIS is responsive to changing impacts of the pandemic. As the data will be gathered through an online survey, it will only include participants with access to the internet. As a result, the sample might be less representative of persons who are illiterate or have limited access to or proficiency with computers or mobile phones. Since none of the sites systematically recruit, there are limitations associated with selective recruitment. As a result, we are unable to generalise the findings to the population of the relevant sites. In addition, the exposure and response rate will vary between countries, making cross-country comparisons challenging. These limitations will be mitigated with the use of multi-level modelling techniques to understand country-wise or cross-cultural predictor variables to obtain meaningful findings. Even though random sampling would be ideal for providing an unbiased representative sample of the overall population, this is not always possible due to resource and time constraints, as the majority of multinational studies relied on non-representative samples. Finally, to minimise participant burden, the survey did not include an assessment of certain relevant variables—such as COVID-19-related stigmatisation, resilience, religious coping or pre-existing mental health conditions.

## Summary

This cross-sectional observational research examines the psychosocial impacts of the COVID-19 pandemic across seven LMICs (Indonesia, Iran, Iraq, Malaysia, Pakistan, Somalia and Turkey) using a novel 32-item CPIS and standardised measures examining well-being, psychological distress, post-traumatic stress and post-traumatic growth. The findings will contribute to the understanding of the psychosocial impacts of the COVID-19 pandemic in different LMICs. In addition, it will provide psychometric data regarding CPIS in the countries of interest. This is a major contribution as there is currently limited access to newly developed scales relating to COVID-19 and associated measures of distress in appropriate languages. As a result, our research will contribute to a growing trend of establishing mental health measures that allow cross-cultural comparisons.

## Ethics and dissemination

Ethical approval was granted by the Human Ethics Committee, University of Otago, New Zealand (Ref. No. 21/102). In addition, international collaborators obtained local authorisation or ethical approval in their respective host universities (Indonesia (Universitas Gadjah Mada Ethics Committee Ref. No. KE/UGM/008/EC/2021); Iraq (Ibn Sina University of Medical and Pharmaceutical Sciences, Institutional Review Board (IRB)—Ethical Committee); Malaysia (Universiti Malaya Research Ethics Committee, Ref. No. UM.TNC 2/UMREC); Pakistan (International Islamic University Islamabad, Institutional Ethical Review Committee, Ref. No. IIUI/ORIC/Ethical-certificate/2021); Somalia (Washington Human Subjects Division (HSD) IRB ID: STUDY00015135) and Turkey (Hasan Kalyoncu Üniversitesi Etik Ethics Committee Ref. No. E-97105791-050.01.01-2342)) before data collection commenced.

Participants will give informed consent to participate in the study before taking part. All data will be fully anonymised and collected and stored securely on the University of Otago, New Zealand Qualtrics platform using an auto-generated non-identifiable letter-number string. Data will be available on reasonable request. Findings will be disseminated by publication in scientific journals and/or conference presentations.

## Author affiliations

[1]Department of Psychological Medicine, University of Otago Christchurch, Christchurch, New Zealand

[2]Faculty of Health, University of Canterbury, Christchurch, New Zealand

[3]School of Clinical Medicine, University of Queensland, Brisbane, Queensland, Australia

[4]Department of Psychological Medicine, University of Otago, Wellington, New Zealand

[5]Department of Anthropology, Australian National University, Canberra, Australian Capital Territory, Australia

[6]Basic Sciences Deptartment, Ibn Sina University of Medical and Pharmaceutical Sciences, Baghdad, Iraq

[7]Department of Social Work, University of Social Welfare and Rehabilitation Science, Tehran, Iran

[8]Department of Psychological Medicine, University of Malaya, Kuala Lumpur, Malaysia

[9]Department of Psychology, Hasan Kalyoncu University, Gaziantep, Turkey

[10]Department of Psychology, International Islamic University, Islamabad, Pakistan

[11]Department of Scholarships and Cultural Relations, Mustansiryah University, Baghdad, Iraq

[12]Lorestan University of Medical Sciences, Khoram-Abad, Iran

[13]Faculty of Economics and Business, Universitas Gadjah Mada, Yogyakarta, Indonesia

[14]Department of Psychology, Islamabad Model College for Girls (PostGraduate), Islamabad, Pakistan

[15]Department of Peace and Conflict Sciences, National Defence University, Islamabad, Pakistan

[16]Faculty of Teacher Training and Education, Universitas Ahmad Dahlan, Yogyakarta, Indonesia

[17]Department of Psychology, University of Washington, Washington, DC, USA

**Acknowledgements** We would like to thank the Research Assistants from the March 15 Project for their work in translating and back-translating study materials.

**Contributors** ST conceived the study, and CB, RS-H, RJP, JB, BB, PJS and SD contributed to the study design. ST, CB and RS-H developed the protocol and selected the measures. ST was responsible for coordinating instrument translation and setting up the online component. ST, WNA-H, MA, ASAN, MD, MJK, MSK, MAMA-M, AM-M, SN, AR, SS, NT, FT, SY, IY and LAZ were responsible for obtaining ethics approval and data acquisition. ST, RB and CB drafted the paper. All authors read, critically revised and approved the final version of the manuscript.

**Funding** The authors have not declared a specific grant for this research from any funding agency in the public, commercial or not-for-profit sectors.

**Competing interests** None declared.

**Patient and public involvement** Patients and/or the public were involved in the design, or conduct, or reporting, or dissemination plans of this research. Refer to the Methods section for further details.

**Patient consent for publication** Not applicable.

**Provenance and peer review** Not commissioned; externally peer reviewed.

## ORCID iDs

Sandila Tanveer http://orcid.org/0000-0002-0648-5382

Philip J Schluter http://orcid.org/0000-0001-6799-6779

Ruqayya Sulaiman-Hill http://orcid.org/0000-0003-0650-1618

Muthana A Majid Al-Masoodi http://orcid.org/0000-0002-1057-7466

Feyza Topçu http://orcid.org/0000-0002-5853-2670

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
