## [Reviewer comments · BMJ Open]

ARTICLE DETAILS

TITLE (PROVISIONAL)	Examining the Psychosocial Impacts of the COVID-19 Pandemic: An International Cross-sectional Study Protocol
AUTHORS	tanveer, sandila; Schluter, Philip; Porter, Richard; Boden, Joseph; Beaglehole, Ben; Sulaiman-Hill, Ruqayya; Dean, Shaystah; Bell, Romana; Al-Hussaini, Wafaa N; Arshi, Maliheh; Amer Nordin, Amer Siddiq; Dinç, Mehmet; Khan, Mussarat; Khoshnami, Mohammad; Majid Al-Masoodi, Muthana A; Moghanibashi-Mansourieh, Amir; Noruzi, Sara; Rahajeng, Anggi; Shaikh, Shaista; Tanveer, Nisa; Topçu, Feyza; Yapan, Saadet; Yuniyanto, Irfan; Zoellner, Lori; Bell, Caroline

VERSION 1 – REVIEW

REVIEWER	EL-Monshed, Ahmed Mansoura University Faculty of Nursing
REVIEW RETURNED	04-Oct-2022

GENERAL COMMENTS	Recommendation: Acceptance with minor modifications Comments: Thank you for the opportunity to review this interesting protocol. These types of studies are needed, and I wish the authors success in the future. Firstly, the authors should follow the general scientific writing guidelines, and the other guidelines of the journal. For example, the authors should follow the available word count according to the journal guidelines. Introduction is clear, but I believe the authors should make a better effort (including with some additional citations) of addressing the similarities/differences in psychosocial impacts of COVID-19 Pandemic globally. This would provide some context for the findings. While such "discussion" should also go in the discussion part of the paper, I am not requesting that the matter be discussed in detail in the introduction, but simply, that some context is given in the introduction to the reader is aware of the "baseline" comparison, prior to delving into the COVID impact. Line 44, page 9, "We developed the Covid Psychosocial Impacts Scale (CPIS) to comprehensively assess the impacts of the pandemic including the adverse and positive personal....). It is better to write about the tool development in method section not in introduction. In method section, give more details about the tool development and describe the tool validity and reliability.
--

REVIEWER	Toure, Almamy Amara University of Conakry
REVIEW RETURNED	30-Nov-2022

GENERAL COMMENTS	General comments This study protocol is expected to reveal low and middle patterns regarding the psychosocial impact of COVID-19; in this regard, it is worth doing. The study integrates different cultures and considers the positive effects of COVID-19, which few explored in previous studies. While the study protocol is well organized, some point needs to be clarified. Introduction The author states, “while others examine a wider range of impacts the pandemic.....social disruption.” From lines 14 to 19. However, no references are mentioned to support the statement. Please insert the references. Methods I am wondering about the value of using two tools (the WHO Well-Being Index (WHO-5) and Kessler-10 (K10)) to recall a recent event that affects people. Could authors give us more explanations for that choice? One of the limitations not mentioned by the authors is that the study does not allow comparison across the time (before and after the pandemic).
--

VERSION 1 – AUTHOR RESPONSE

Reviewer: 1

General Comments

Thank you for the opportunity to review this interesting protocol. These types of studies are needed, and I wish the authors success in the future.

We thank Reviewer 1 for providing us with this positive feedback. In particular, for acknowledging that it is a very interesting study, and recognising the need. We greatly appreciate these remarks and we would also like to thank them for wishing us future success.

Firstly, the authors should follow the general scientific writing guidelines, and the other guidelines of the journal. For example, the authors should follow the available word count according to the journal guidelines.

When drafting the manuscript we consulted the journal's guidelines for protocol papers. From our reading and understanding of these guidelines, papers must not exceed 4,000 words (excluding the title page, abstract, references, figures and tables). According to the default MS Word Count function, our paper totals 3,478 words, consistent with these guidelines. We hope this addresses Reviewer 1's concerns.

Introduction is clear, but I believe the authors should make a better effort (including with some additional citations) of addressing the similarities/differences in psychosocial impacts of COVID-19 Pandemic globally. This would provide some context for the findings. While such "discussion" should also go in the discussion part of the paper, I am not requesting that the matter be discussed in detail in the introduction, but simply, that some context is given in the introduction to the reader is aware of the "baseline" comparison, prior to delving into the COVID impact.

As suggested, we have now provided further context and additional citations providing some context in the Introduction section (pages 6 and 7). In addition, the Discussion section has been updated to address the similarities/differences between the current design with existing multi-national studies assessing the psychosocial impacts of the COVID-19 pandemic (pages 13 and 14).

Line 44, page 9, “We developed the Covid Psychosocial Impacts Scale (CPIS) to comprehensively assess the impacts of the pandemic including the adverse and positive personal....). It is better to write about the tool development in method section not in introduction. In method section, give more details about the tool development and describe the tool validity and reliability.

We considered shifting the information about tool development to the Methods section as suggested. However, the present study describes the international use of the CPIS as opposed to its development. We, therefore, believe that the details of tool development most appropriately belong in the Introduction section and have therefore largely left this section unchanged. The findings of the psychometric study examining the reliability and validity of the CPIS in a New Zealand population have now been added briefly in the Introduction section (pages 7 and 8).

Reviewer: 2

General Comments

This study protocol is expected to reveal low and middle patterns regarding the psychosocial impact of COVID-19; in this regard, it is worth doing. The study integrates different cultures and considers the positive effects of COVID-19, which few explored in previous studies. While the study protocol is well organized, some points need to be clarified.

We also thank Reviewer 2 for their very positive appraisal, recognition of the importance of this work, and for describing it as well organized.

Introduction

The author states, "while others examine a wider range of impacts the pandemic.....social disruption." From lines 14 to 19. However, no references are mentioned to support the statement. Please insert the references.

Reviewer 2 requested references to support the statement "while others examine a wider range of impacts the pandemic.....social disruption." Thank you for highlighting this. We have now provided reference of 'The CoRonavlrus health Impact Survey (2020)' in support of our statement (page 6).

Methods

I am wondering about the value of using two tools (the WHO Well-Being Index (WHO-5) and Kessler-10 (K10)) to recall a recent event that affects people. Could authors give us more explanations for that choice?

Reviewer 2 requested more explanation on the choice of the WHO Well-Being Index (WHO-5) and Kessler-10 (K10) to recall a recent event that affects people. Regarding the assessment timeline, we have adhered to the standard instructions of the tools. Only a few of the instructions have been modified to account for COVID-19 as a potentially traumatic event.

We have included these standardised measures to examine the construct validity of novel CPIS as our previous work indicates that CPIS correlates with psychological distress and general wellbeing (page 8). This has now been mentioned under the heading 'Psychometric analyses' in the Methods section (page 12).

One of the limitations not mentioned by the authors is that the study does not allow comparison across the time (before and after the pandemic).

Reviewer 2 noted an additional limitation in that our study does not allow comparison across the time (before and after the pandemic). We thank the reviewer for this suggestion and we have now included this as a limitation (page 14).

We hope this letter and the revised manuscript adequately address the reviewers' comments. All authors agree with the changes made and have approved this revised manuscript. If there are additional points requiring clarification or if more information is required, please do not hesitate to contact me.